# Modulation of the *ETV6::RUNX1* Gene Fusion Prevalence in Newborns by Corticosteroid Use During Pregnancy

**DOI:** 10.3390/ijms26072971

**Published:** 2025-03-25

**Authors:** Leticia Benítez, Ute Fischer, Fàtima Crispi, Sara Castro-Barquero, Francesca Crovetto, Marta Larroya, Lina Youssef, Ersen Kameri, Helena Castillo, Clara Bueno, Rosa Casas, Roger Borras, Eduard Vieta, Ramon Estruch, Pablo Menéndez, Arndt Borkhardt, Eduard Gratacós

**Affiliations:** 1BCNatal|Fetal Medicine Research Center, Hospital Clínic and Hospital Sant Joan de Deu, University of Barcelona, 08950 Barcelona, Spain; fcrispi@clinic.cat (F.C.); sacastro@clinic.cat (S.C.-B.); francesca.crovetto@sjd.es (F.C.); larroya@clinic.cat (M.L.); lyoussef@recerca.clinic.cat (L.Y.); hcastillo@clinic.cat (H.C.); egratacos@ub.edu (E.G.); 2Institut d’Investigacions Biomèdiques August Pi i Sunyer, 08036 Barcelona, Spain; 3Department of Pediatric Oncology, Hematology and Clinical Immunology, University Children’s Hospital, Medical Faculty, Heinrich Heine University, 40225 Düsseldorf, Germany; ute.fischer@med.uni-duesseldorf.de (U.F.); ersen.kameri@med.uni-duesseldorf.de (E.K.); 4German Cancer Consortium, Partner Site Essen-Düsseldorf, 40225 Düsseldorf, Germany; 5Deutsches Krebsforschungszentrum (DKFZ), 69120 Heidelberg, Germany; 6Red Española de Terapias Avanzadas—Instituto de Salud Carlos III (ISCII), 28029 Madrid, Spain; 7Centro de Investigación Biomédica en Red Cáncer(CIBER-ONC), Instituto de Salud Carlos III (ISCIII), 28029 Madrid, Spain; 8Institut de Recerca Sant Joan de Déu, 08950 Barcelona, Spain; 9Josep Carreras Leukemia Research Institute, 08916 Barcelona, Spain; cbueno@carrerasresearch.org (C.B.); pmenendez@carrerasresearch.org (P.M.); 10Department of Biomedicine, School of Medicine, University of Barcelona, 08014 Barcelona, Spain; restruch@clinic.cat; 11Institució Catalana de Recerca i Estudis Avançats, 08010 Barcelona, Spain; 12Centro de Investigación Biomédica en Red de Fisiopatología de la Obesidad y Nutrición, 28029 Madrid, Spain; rcasas1@recerca.clinic.cat; 13Institut de Recerca en Nutrició i Seguretat Alimentaria, University of Barcelona, 08014 Barcelona, Spain; 14Department of Internal Medicine, Hospital Clinic, University of Barcelona, 08014 Barcelona, Spain; rborras@recerca.clinic.cat; 15Centro de Investigación Biomédica en Red y Salud Mental, CIBERSAM, Instituto de Salud Carlos III, 28029 Madrid, Spain; evieta@clinic.cat; 16Department of Psychiatry and Psychology, Hospital Clinic, Neuroscience Institute, IDIBAPS, University of Barcelona, CIBERSAM, 08014 Barcelona, Spain

**Keywords:** *ETV6::RUNX1*, childhood leukemia, cord blood, prenatal, corticosteroids

## Abstract

*ETV6::RUNX1*-positive pediatric acute lymphoblastic leukemia frequently has a prenatal origin and follows a two-hit model: a first somatic alteration leads to the formation of the oncogenic fusion gene *ETV6::RUNX1* and the generation of a preleukemic clone in utero. Secondary hits after birth are necessary to convert the preleukemic clone into clinically overt leukemia. However, prenatal factors triggering the first hit have not yet been determined. Here, we explore the influence of maternal factors during pregnancy on the prevalence of the *ETV6::RUNX1* fusion. To this end, we employed a nested interventional cohort study (IMPACT-BCN trial), including 1221 pregnancies (randomized into usual care, a Mediterranean diet, or mindfulness-based stress reduction) and determined the prevalence of the fusion gene in the DNA of cord blood samples at delivery (*n* = 741) using the state-of-the-art GIPFEL (genomic inverse PCR for exploration of ligated breakpoints) technique. A total of 6.5% (*n* = 48 of 741) of healthy newborns tested positive for *ETV6::RUNX1*. Our multiple regression analyses showed a trend toward lower *ETV6::RUNX1* prevalence in offspring of the high-adherence intervention groups. Strikingly, corticosteroid use for lung maturation during pregnancy was significantly associated with *ETV6::RUNX1* (adjusted OR 3.9, 95% CI 1.6–9.8) in 39 neonates, particularly if applied before 26 weeks of gestation (OR 7.7, 95% CI 1.08–50) or if betamethasone (OR 4.0, 95% CI 1.4–11.3) was used. Prenatal exposure to corticosteroids within a critical time window may therefore increase the risk of developing *ETV6::RUNX1*+ preleukemic clones and potentially leukemia after birth. Taken together, this study indicates that *ETV6::RUNX1* preleukemia prevalence may be modulated and potentially prevented.

## 1. Introduction

Childhood B-cell acute lymphoblastic leukemia (B-ALL) is the most frequent pediatric cancer and one of the leading causes of childhood mortality in developed countries, with increasing incidence in the past years [1,2,3]. Although advances in molecular risk-stratification, diagnosis, and targeted treatments have significantly improved survival rates, B-ALL remains a health priority that fits the standards for primary prevention [4,5,6]. 

The most prevalent subtypes of B-ALL frequently have a prenatal origin, following a two-hit model: (1) Somatic acquisition of oncogenic fusion genes (such as *ETV6::RUNX1* or *TCF3::PBX1*) or aberrant chromosome numbers (e.g., high hyperdiploidy) occurs in hematopoietic stem or progenitor cells as a first genetic alteration arising during prenatal life. (2) Secondary oncogenic hits after birth convert the preleukemic clone into clinically overt leukemia [7,8,9]. Retrospective analyses of Guthrie cards from children who subsequently developed leukemia demonstrated the presence of such first-hit molecular lesions at birth [10,11]. Such molecular lesions have subsequently been detected in healthy newborns [12,13,14].

The reported prevalence of the *ETV6::RUNX1* fusion was about 100× higher than the actual leukemia incidence, indicating that the fusion gene is only mildly oncogeneic.

In twins with concordant *ETV6::RUNX1*-positive leukemia, ALL develops at different times, and postnatal latency can be protracted [15]. Most frequently, *ETV6::RUNX1*+ B-ALL harbors deletions of the second, non-translocated ETV6 allele [16], implying that the initial gene fusion predisposes individuals to leukemia, while subsequent ETV6 deletion acts as a promoting factor. While the postnatal triggers of leukemia have been extensively investigated [8], few studies have focused on the association between prenatal factors and preleukemic lesions. Parental age, ethnicity, maternal diet, folate intake, alcohol consumption, X-ray exposure, pesticides, perinatal infections, and fetal growth may play a role [17].

The investigation of prenatal determinants of childhood leukemia is limited by the technical challenge of evaluating a wide spectrum of preleukemic lesions in large sample sizes [18]. Cord blood (CB) biobanks from well-characterized large cohorts would be the most obvious source for this research, but sample sizes are commonly small and rarely suitable for this type of investigation. First, the reported frequency of cells in healthy newborns with preleukemic lesions at birth is estimated to range from 1 in 10^3^ to 10^4^, making them undetectable by most available laboratory methods. Moreover, the detection of these lesions has been optimized for RNA rather than DNA owing to patients’ unique genomic DNA breakpoints, often occurring in extensive genomic regions not targetable by conventional techniques. Unfortunately, RNA samples are very rarely available in CB biobanks, are very unstable, and are susceptible to contamination, making them non-amenable for these studies.

Fueller et al. recently designed a new laboratory technique called “Genomic inverse PCR for exploration of ligated breakpoints” (GIPFEL) [19], which allows for the sensitive detection of recurrent chromosomal translocations. This technique relies on PCR-based quantification of unique DNA sequences that are created by circular ligation of restricted genomic DNA from translocation-bearing cells. Therefore, prior knowledge of the individual-specific interchromosomal breakpoints is not required. Recently, GIPFEL analysis allowed the assessment of the prevalence of preleukemic lesions in a Danish cohort of 1000 CBs from healthy newborns, and yielded a prevalence of 5% for *ETV6::RUNX1* and 0.6% for *TCF3::PBX1* [20,21].

Previous studies evaluating prenatal risk factors for childhood leukemia were mostly retrospective case-control studies, and CB samples were rarely available [17]. However, in the IMPACT-BCN (Improving Mothers for a better PrenAtal Care Trial BarCeloNa) randomized trial, CB samples were collected at delivery, and CB-derived mononuclear cells were stored as part of the study design. The study evaluated the impact of lifestyle interventions (Mediterranean diet and stress reduction) in 1221 pregnant women and demonstrated benefits on several perinatal outcomes [22].

In the present study, we evaluated the association between prenatal interventions, maternal and perinatal factors, and the emergence of neonatal preleukemic *ETV6::RUNX1* lesions in the well-characterized cohort of CB samples from the IMPACT-BCN trial.

## 2. Results

### 2.1. High Adherence to Mediterranean Diet and Stress Reduction Intervention Is Associated with a Trend Toward Lower Prevalence of the ETV6::RUNX1 Fusion

The *ETV6::RUNX1* fusion gene was screened using GIPFEL in *n* = 741 cord blood samples (Figure 1) in a blinded fashion (*n* = 245 without intervention, *n* = 250 Mediterranean diet intervention, and *n* = 246 stress reduction intervention). The *ETV6::RUNX1* fusion was detected in 48 cord blood samples (6.5%) of the study population, as determined by GIPFEL analysis (Appendix A).

High adherence to the Mediterranean diet intervention was defined as an improvement of at least 3 points in the final score of the 17-item dietary screener compared with the baseline score. Adherence to the stress reduction intervention was considered high if at least 6 of 9 stress reduction sessions were attended.

Regarding the intervention group, there were no differences in the prevalence of neonatal *ETV6::RUNX1* fusion gene among the three groups. However, when only considering participants who demonstrated a high adherence to the intervention, a non-significant trend toward lower neonatal ETV6::RUNX1 prevalence was detected in both interventions (prevalence of 7.0% for usual care versus 5.9% for stress reduction (OR 0.8; 95% CI 0.3–2.0) versus 5.3% for Mediterranean diet (OR 0.7; 95% CI 0.3–1.7) (Figure 2A, Table 1).

### 2.2. Univariate Analysis Identifies Corticosteroid Application During Pregnancy as Significantly Associated with the Presence of ETV6::RUNX1 Fusions

Maternal and pregnancy characteristics of the study population were evaluated using univariate analyses based on the presence of the neonatal *ETV6::RUNX1* fusion gene (Table 2 and Appendix A). Considering baseline characteristics, univariate analysis identified Maghreb ethnicity (OR 12.8, *p* < 0.001) and study class (OR for secondary/technological studies compared to primary/no studies 0.3, *p* = 0.046; OR for university studies compared to primary/no studies 0.7, *p* = 0.525) as significantly associated with the appearance of preleukemic lesions. Among perinatal characteristics, the administration of exogenous corticosteroids during pregnancy was significantly associated with the preleukemic status (OR 3.5, *p* = 0.005) (Figure 2B), while folate supplementation showed a non-significant preventive tendency (OR 0.6, *p* = 0.056). No specific dietary component or key food was identified as significantly associated with the neonatal *ETV6::RUNX1* fusion gene (Appendix A).

### 2.3. Multivariate Regression Models Report a Higher Probability of ETV6::RUNX1 Occurrence for Corticosteroid Administration Before 26 Weeks of Gestation and Use of Betamethasone

Multivariate regression analysis confirmed the administration of exogenous corticosteroids as significantly associated with the *ETV6::RUNX1* fusion gene (Table 3).

Regarding the use of exogenous corticosteroids, a subanalysis according to gestational age reported a higher estimated probability in those participants receiving corticosteroids before 26 weeks of gestation (estimated probability of neonatal *ETV6::RUNX1* appearance of 0.50 (95% CI 0.1–0.9)) compared to those who received the treatment over 26 weeks of pregnancy (estimated probability 0.11 (95% CI 0.01–0.23); OR for administration before 26 weeks 7.7 (95% CI 1.08–50), *p* = 0.04). Betamethasone (*n* = 25) was the corticosteroid class most commonly used (with most participants receiving 2 doses for fetal lung maturation purposes), followed by methylprednisone (*n* = 1), prednisone (*n* = 11), and topical use (*n* = 2). GR:MR refers to the glucocorticoid receptor (GR) to mineralocorticoid receptor (MR) activity ratio for each exogenous corticosteroid listed (Table 4 and Appendix A).

## 3. Discussion

This study identified prenatal exposure to corticosteroids as a potential risk factor for prenatal *ETV6::RUNX1* fusions and, thus, for leukemia development later in life. In addition, Maghreb ethnicity was associated with *ETV6::RUNX1* fusions, although absolute numbers were small. There were no other statistically significant associations between maternal lifestyle interventions and the prevalence of preleukemic *ETV6::RUNX1* lesions revealed in our study.

Previous studies have evaluated the association of prenatal risk factors [23,24,25,26,27,28,29,30,31,32,33,34] that suggest maternal age, ethnicity, some dietary components, folate supplementation, alcohol consumption, X-ray exposure, pesticide exposure, perinatal infections, and fetal growth as probable risk factors. Nevertheless, most available evidence is based on retrospective case-control studies that used self-reported questionnaires, which are susceptible to recall and underreporting bias. These studies were also unable to differentiate prenatal from postnatal exposures for most risk factors.

Our current study adds to previous evidence by reporting the analysis of 741 cord blood samples from a well-phenotyped cohort of pregnant women included in the IMPACT-BCN randomized trial. Maternal and prenatal factors were determined prospectively during the trial by the use of medical records by a trained medical doctor, avoiding recall bias related to self-report [35]. Additionally, assessment of maternal diet was prospectively performed by an expert nutritionist in a prospective manner during pregnancy, with proven validity [31]. Likewise, the use of biomarkers to confirm adherence to interventions enhances the reliability of results. The prevalence of *ETV6::RUNX1* was large enough to allow statistical comparisons.

The neonatal *ETV6::RUNX1* positivity prevalence of 6.5% (48 out of 741) is in line with previously published papers on healthy newborns in the USA, Europe, and Japan, ranging from 0.01 to 7% [14].

There was a significant association between maternal Maghreb ethnicity and neonatal preleukemic fusion genes. Although differences in the incidence of childhood leukemia between ethnicities had been previously described, an association with Maghreb ethnicity has not been reported. Concerning the *ETV6::RUNX1* fusion gene, Western countries and Hispanics have consistently described as presenting higher rates [36] compared with Black and Asian ethnicities. Genomic profiles [37] and folate metabolism polymorphisms [4] have been hypothesized as underlying factors. The difference in Maghreb ethnicity could be explained by genetic polymorphisms or by lifestyle or environmental determinants not recorded during the randomized trial. However, the small number of samples hampered the analysis of dietary components in this specific subpopulation.

Regarding corticosteroids, the main indications for their use in our cohort were fetal lung maturation in cases at risk of preterm birth and maternal autoimmune or dermatological conditions. To address the potential confounding effect of the indication for steroid administration, we conducted additional analyses adjusting for clinical conditions that warranted steroid use. These adjustments confirmed that the association between corticosteroid exposure and *ETV6::RUNX1* positivity remained significant, independent of the underlying indication for treatment. This suggests that the effect is more likely attributable to the drug itself rather than the condition requiring its administration. Prenatally, corticosteroids improve fetal lung maturation but have strong effects on fetal somatic and brain growth. This study identified exogenous corticosteroids as risk factors for preleukemic *ETV6::RUNX1* lesions. To our knowledge, this association has not previously been reported. Most cases had two doses of betamethasone under 26 weeks of gestation. In fact, glucocorticoids represent a key drug in the treatment backbone of B-ALL [38]. However, frequent secondary lesions in *ETV6::RUNX1*-positive ALL may directly affect the glucocorticoid response (e.g., by mutation of the glucocorticoid receptor NR3C1), the cell death pathway induced via its signaling, or components of the mismatch repair pathways [39]. One speculation on the effect of glucocorticoid treatment on the increased incidence of *ETV6::RUNX1* lesions in our study may therefore be an expansion of pre-existing minor *ETV6::RUNX1*-positive clones, due to secondary mutations that arise under the selective pressure of betamethasone. In this case, children would be expected to have a higher likelihood of developing leukemia later in life, which needs to be evaluated in larger studies with long-term follow-up.

Another probable explanation is the generation of the fusion gene itself as a direct consequence of glucocorticoid treatment. Recently, physiological concentrations of glucocorticoids have been shown to induce DNA double-strand breaks via inhibition of topoisomerase II. The effect of different glucocorticoids depended on their affinity for the glucocorticoid receptors (Figure 3) [40]. Betamethasone was significantly associated with an increased incidence of ETV6::RUNX1 fusions in our study and has a higher affinity for the receptor compared to dexamethasone, prednisolone, or endogenous cortisol [40]. Repair of the DNA double-strand breaks by non-homologous end joining could lead to *ETV6::RUNX1* fusion. Interestingly, after glucocorticoid treatment of B-ALL patients, increased numbers of *ETV6::RUNX1*-positive lymphocytes have been detected [41].

As recently demonstrated by Kim et al., using bisulfite sequencing of DNA extracted from bloodspot “Guthrie” cards, glucocorticoids also induce genome-wide methylation differences following antenatal exposure [43]. Methylation changes have been associated with prenatal environmental exposures and an increased risk of developing childhood leukemia [44], and there is also a correlation of methylation changes and specific genetic subtypes of ALL [45]. Hypomethylated areas are likely transcriptionally active regions and have a more open chromatin that may lend itself to interchromosomal contacts, DNA breaks, and aberrant repair [44]. It is tempting to speculate that the ETV6 and RUNX1 genes that recombine with one another so frequently are spatially localized together in transcriptional or chromatin compartments. The clustering of breakpoints, as seen in ETV6::RUNX1-positive ALL, also likely reflects selection for functional rearrangements during tumorigenesis. Although data from leukemia is lacking, previous studies in prostate cancer [46,47] show a significantly reduced methylation at breakpoints of specific interchromosomal translocations.

Glucocorticoid administration during pregnancy may also lead to permanent changes that impact on the risk of developing leukemia after birth (Figure 4). Glucocorticoid administration during pregnancy leads to higher plasma cortisol levels throughout adult life, indicating persistent reprogramming of the hypothalamus–pituitary–adrenal axis in animal models [48]. In the adrenal hypothesis model proposed by Schmiegelow et al. [49], early childhood infections result in profound changes in the hypothalamus–pituitary–adrenal axis that increase plasma cortisol levels and protect from leukemia by direct elimination of preleukemic cells via immune modulation and promotion of Th1-cytokine response [49]. As a possible explanation, it is conceivable that glucocorticoids may work as a double-edged sword: They may increase the risk of preleukemic fusions arising in utero but may also reduce the risk of leukemic transformation after birth.

Our data indicate that the effects of glucocorticoids in early pregnancy and childhood need to be carefully studied in larger cohorts, and the pros and cons of their application need to be deliberated by informed clinicians. Additionally, our study indicates promising options for potential strategies to prevent leukemia in childhood.

A previous study has reported an association between maternal diet quality and the risk of childhood leukemia [23]. In the present study, no significant associations were found between the Mediterranean diet or specific dietary components and the rate of *ETV6::RUNX1*. However, there was a trend toward lower rates among pregnancies with a high adherence to the Mediterranean diet, which deserves further investigation in larger sample sizes. Foods containing DNA topoisomerase II inhibitors, including coffee, tea, and other caffeinated beverages, canned food or dried vegetables or legumes, cocoa, red wine, apples, and berries, have been described as potential childhood leukemia triggers [50,51]. Concerning vitamins, deficient maternal folate intake during pregnancy has been extensively described as a risk factor for ALL [4,18,24,27,52,53]. In the present study, there was a non-significant trend toward lower rates of *ETV6::RUNX1* in relation to folate intake, although we cannot separate the intrinsic effect of other vitamins that are commonly taken for supplementation during pregnancy. There was no association between smoking, alcohol, or other toxics and *ETV6::RUNX1* fusion gene, but the prevalence of these risk factors was too low to allow meaningful comparisons. A lower socioeconomic class has been suggested to have a preventive effect on childhood cancer postnatally [54,55]. In this study, we observed a non-significant trend toward the opposite effect on *ETV6::RUNX1* frequency, which could be related to a lower folate intake in this subgroup of women.

Some limitations of our study should be acknowledged. First, the fact that the population of the study was from a single center located in Spain, which, although including several ethnicities, could hamper the external validation of results. Second, given the small sample size of *ETV6::RUNX1* positive individuals, our findings should be considered preliminary and require validation in larger studies to confirm their biological and clinical significance. Third, the variability in reported *ETV6::RUNX1* prevalence in cord blood across studies may stem from differences in detection methods, sample sizes, and population characteristics; however, the rigorous methodology and quality controls in our study support the credibility of our findings within this context. Fourth, we performed a 4-year follow-up in which none of the *ETV6::RUNX1*-positive newborns have been diagnosed with acute lymphoblastic leukemia. However, given the relatively short follow-up period, we acknowledge that longer-term monitoring is needed, and child follow-up should be warranted to confirm or discard the development of overt leukemia.

## 4. Material and Methods

### 4.1. Study Design and Participant Selection

This study represents a secondary analysis of the IMPACT-BCN trial, a randomized clinical trial conducted at BCNatal (Hospital Clínic and Hospital Sant Joan de Deu, Barcelona, Spain) from 2017 to 2020 [56]. The trial was approved by the Institutional Review Board (HCB-2016-0830). All participants provided written informed consent. Participants were screened for eligibility during routine second-trimester ultrasound scans (19–23.6 weeks of gestation) for being at high risk of developing small-for-gestational-age newborns [57].

### 4.2. Interventions

Participants were randomly assigned 1:1:1 to one of the three study groups: a Mediterranean diet intervention, a stress reduction program intervention, or usual care without any additional intervention (control group).

The Mediterranean diet intervention was adapted from the PREDIMED trial [58]. Registered dietitians conducted group meetings, as well as monthly face-to-face and additional telephone interviews to ensure adherence to the diet. All participants received olive oil (2 L every month) and 15 g of walnuts per day (450 g every month) at no cost. The participant received dietary training to encourage an increased intake of whole grain cereals (≥5 servings/d); vegetables and dairy products (≥3 servings/d); fresh fruit (≥2 servings/d); and legumes, nuts, fish, and white meat (≥3 servings/week). A 151-item food frequency questionnaire, a 7-day dietary journal, and a 17-item dietary assessment score (range, 0–17) were used to assess baseline nutrients and vitamin intake of all participants and adherence to the Mediterranean diet for the intervention group. A high adherence was defined as an improvement of at least 3 points in the final score of the 17-item dietary screener compared with the baseline score.

Led by experienced, certified instructors, the stress reduction group received an 8-week-long program of structured intervention (based on mindfulness) tested in clinical trials [59]. It included weekly 2.5-h sessions, 1 full-day session, and daily home practice, formal and informal techniques, with the goal of enhancing awareness and reducing anxiety. The sessions included didactic presentations, formal 45-min meditation practices, yoga, body awareness, and group discussions. Adherence to the stress reduction intervention was considered high if at least 6 of 9 stress reduction sessions were attended.

A detailed description of the interventions was previously published [22,56].

### 4.3. Cord Blood Sample Collection

Cord blood samples were collected at delivery after cord clamping, usually between 30 s to 1 min following international recommendations for delivery, and immediately stored at 4 °C. Ficoll density-gradient centrifugation (GE Healthcare, headquartered in Chicago, IL, USA) was performed in the following hours by an expert technician who was blinded to the intervention group. Cord blood components (serum, plasma, and mononuclear cells) were stored separately at −80 °C. Mononuclear cells were shipped to the University Hospital Düsseldorf, Department of Pediatric Oncology, Hematology, and Clinical Immunology for detection of *ETV6::RUNX1* fusions by GIPFEL analysis.

### 4.4. Genomic Inverse PCR for Exploration of Ligated Breakpoints (GIPFEL)

Cryopreserved cord blood samples were screened using the GIPFEL technique to check the neonatal *ETV6::RUNX1* positivity as detailed in the Appendix A and Methods. In brief, genomic DNA was isolated from cord blood samples and digested with the restriction enzyme SacI. After circularization mediated by T4 DNA ligase, ETV6 and RUNX1 joints are ligated if the translocation is present. Circularized/ligated DNA was amplified, and *ETV6::RUNX1* was detected by real-time PCR, agarose gel electrophoresis, and confirmed by Sanger sequencing. Researchers and technicians performing the laboratory analysis were blinded to the intervention group.

### 4.5. Predictive Variables

The main outcome was the presence of the neonatal *ETV6::RUNX1* fusion gene in CB samples. All maternal and prenatal variables were considered potential predictive variables and were obtained prospectively at the beginning or during the trial, including maternal age, body mass index, ethnicity, level of education, socioeconomic status, parity, medical conditions, cigarette smoking, alcohol or recreational drug intake during pregnancy, folate supplementation, use of exogenous corticosteroids, occurrence of pregnancy complications (gestational diabetes, preterm birth, preeclampsia, small-for-gestational-age), and gestational age at delivery. Ethnicity was self-reported by participants among White, Latin, Afro-American, Maghreb, or others. Maghreb ethnicity was defined as patients from Western and Central North Africa. Study class was divided into primary/no studies, secondary/technology, and university studies and was self-reported by participants. Socioeconomic status was defined as low if participants reported having never worked or being unemployed for more than two years and having a partner with unskilled work or who was unemployed; and high if they reported university studies regardless of whether they were working; and medium if any other situations. Exogenous use of corticosteroids was determined according to medical records, which specified the dose, class, and duration of use. Gestational diabetes was defined as any degree of glucose intolerance with onset during pregnancy, in our setting diagnosed by the presence of two altered values in a 100-g, 3-h oral glucose tolerance test. Preterm birth was defined as delivery before 37 weeks of gestation. Preeclampsia was defined as the concurrence of pregnancy-onset hypertension and proteinuria. Small-for-gestational-age was defined as birthweight below the 10th centile according to local standards. Steroid use included betamethasone for preterm labor and oral or topical prednisone for maternal autoimmune and skin conditions that were administered during the entire pregnancy. Maternal dietary key food and nutrient intake were obtained by a specialized nutritionist at the beginning and end of the intervention with several questionnaires validated for the present study population [60].

### 4.6. Statistical Analysis

Maternal and prenatal data are presented as mean (standard deviation, SD) or number (percentage), as appropriate. Statistical analysis for comparison of clinical and prenatal characteristics included the use of the Student’s *t*-test for continuous variables, the Chi-square test for categorical variables, and logistic univariate and multivariate regression analyses. Data were expressed using odds ratios and their corresponding 95% confidence intervals. Multivariate regression models were adjusted for significant variables in our study and described confounders. Estimated probability (or estimated marginal means) was computed with emmeans library version 1.8.6. Differences were considered significant when the *p*-value was lower than 0.05. Statistical analyses were conducted using Stata 15.1 (StataCorp LP, Houston, TX, USA) and R version 4.2.1 (Vienna, Austria).

## 5. Conclusions

Prenatal exposure to corticosteroids within a critical time window may therefore increase the risk of developing *ETV6::RUNX1+* preleukemic clones and potentially leukemia after birth. Taken together, this study indicates that the prevalence of *ETV6::RUNX1* preleukemia may be modulated and potentially prevented.

## Figures and Tables

**Figure 1 ijms-26-02971-f001:**
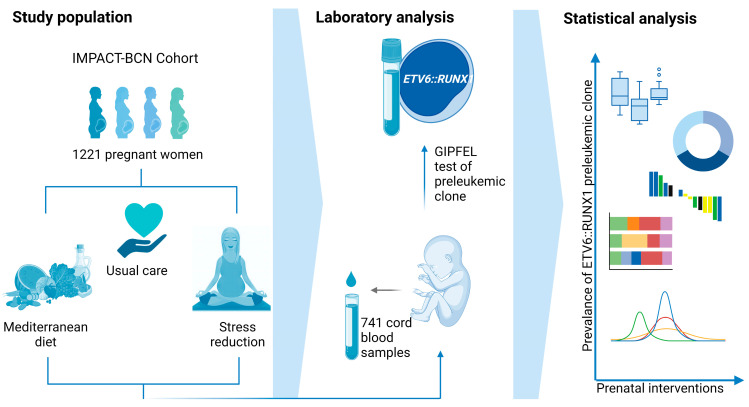
Study design depicting the study population, the intervention arms, and performed laboratory and statistical analyses. In the nested interventional cohort study (IMPACT-BCN trial).

**Figure 2 ijms-26-02971-f002:**
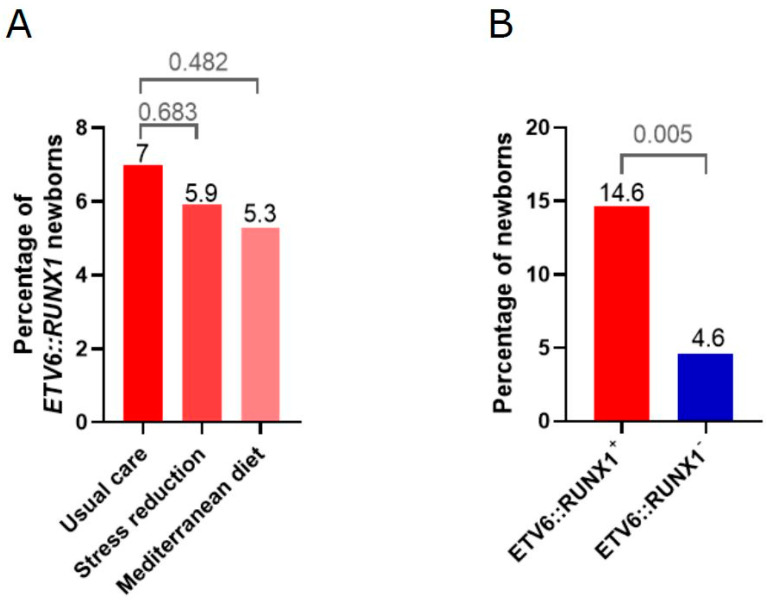
High adherence to a Mediterranean diet and stress reduction interventions, as well as exogenous glucocorticoid application, is associated with a lower prevalence of the *ETV6::RUNX1* fusion as determined by the GIPFEL method. (**A**) The bar graphs present the percentage of *ETV6::RUNX1* fusions considering only participants with high adherence to the interventions showing a non-significant trend toward lower prevalence (stress reduction (OR = 0.8, *p* = 0.683) and mediterranean diet (OR = 0.7, *p* = 0.482)). (**B**) The bar graph presents the prevalence of *ETV6::RUNX1* in newborns whose mothers received exogenous corticosteroids during pregnancy (*n* = 39 in total; *ETV6::RUNX1*+ *n* = 7 of 48; *ETV6::RUNX1*− *n* = 32 of 693; OR = 3.5, *p* = 0.005).

**Figure 3 ijms-26-02971-f003:**
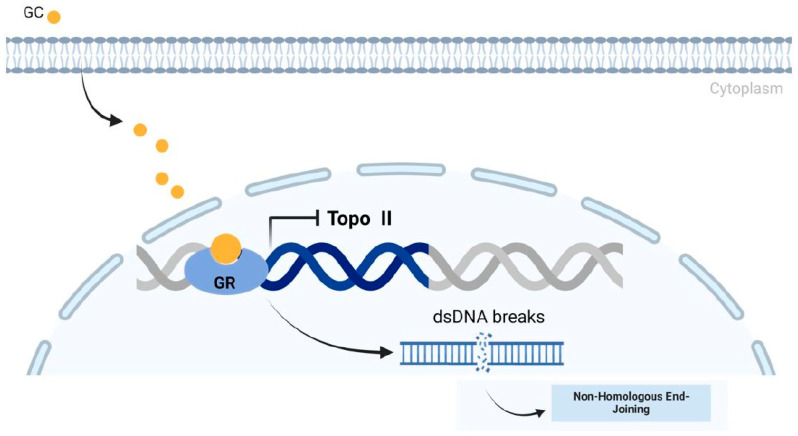
Glucocorticoids can inhibit DNA topoisomerase II (Topo II) activity, potentially by downregulating its expression or modifying its access to DNA through chromatin remodeling. Reduced Topo II activity compromises its role in resolving DNA supercoiling, untangling chromatids, and repairing double-strand breaks (DSBs). One proposed mechanism for leukemia-causing chromosomal translocations entails chromosomal breakage by DNA topoisomerase II and recombination of DNA free ends from different chromosomes through DNA repair. Topo II inhibition increases the risk of chromosomal translocations, such as those involving the *MLL* gene rearrangements on chromosome 11q23, a hallmark of de novo acute leukemia development. Similarly, dysregulated Topo II activity has been linked to secondary leukemias, particularly therapy-related acute myeloid leukemia (t-AML) [42].

**Figure 4 ijms-26-02971-f004:**
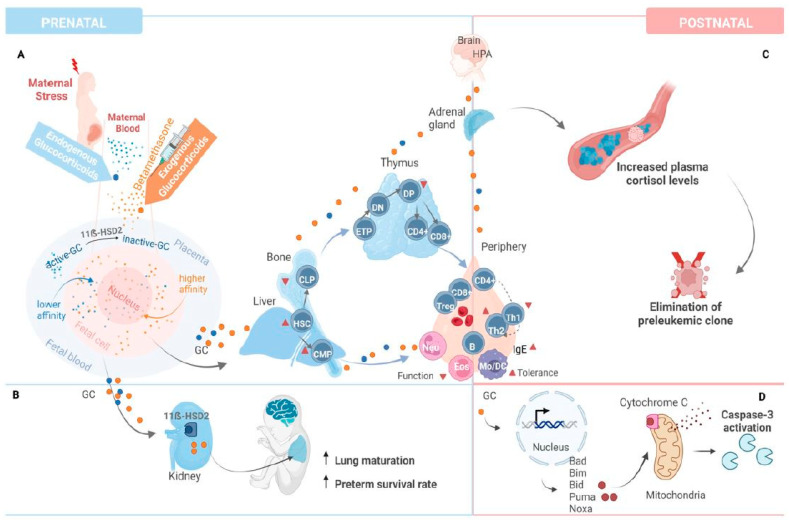
Potential impact of glucocorticoids on fetal prenatal and early postnatal development, as well as ALL therapy. (**A**) Endogenous glucocorticoid (cortisol) levels may increase due to prenatal stress perception during the second trimester. However, only a limited amount of endogenous GC crosses the placental barrier as they serve as a good substrate for the placental 11β-HSD2 enzyme, getting inactivated. Hence, exogenous GCs, such as betamethasone, may be administered to promote fetal lung maturation. These GCs, being steroid compounds, can easily cross the placenta and bind to the glucocorticoid receptor (GR) with a higher affinity compared to endogenous ones. Particularly, betamethasone shows a higher affinity for GR than other GCs. Upon binding, GR translocates to the nucleus, where it acts as either a transcriptional activator or repressor of target genes. Fetal exposure to high levels of GCs, whether from elevated maternal stress or medical treatment, may promote myeloid hematopoiesis and bone marrow or bone erythropoiesis by shifting hematopoietic stem cell (HSC) differentiation toward common myeloid progenitors (CMP) rather than common lymphoid progenitors (CLP). Additionally, GCs can influence bone marrow stromal cells, such as osteoblasts, which release soluble factors that regulate HSC differentiation and proliferation. Such changes in hematopoiesis may correlate with compromised humoral (B cell-derived) immune responses and perinatal neutrophil (Neu) function. GC excess in the thymus acts as a potent inducer of immature double-positive (DP) thymocyte apoptosis, as GCs are, in addition, locally produced here toward the end of the pregnancy, subsequently accelerating the maturation of double-negative (DN) thymocytes to occupy the available niche. By inhibiting T-cell receptor signaling (TCR) or autoimmune regulator (AIRE)-mediated autoantigen transcription, glucocorticoids may mitigate the apoptosis event, allowing autoreactive CD4 and CD8 single-positive (SP) T cells to circulate. Prenatal GC exposure also programs CD4 T helper (Th) cells toward a Th2 response. Furthermore, prenatal glucocorticoids may increase postnatal HPA axis activity, resulting in elevated levels of corticotropin-releasing hormone (CRH) and arginine vasopressin (AVP). This hyperactivity can enhance both innate and adaptive immune responses, potentially leading to monocyte (Mo), macrophage, and dendritic cell (DC) tolerance to pathogens or excessive mast cell degranulation. These prenatal adaptations in immune function may increase the risk of infections, asthma, and other immune-related disorders in later life. (**B**) Exogenous GC, in contrast to endogenous GC, can bypass the placental and fetal 11ẞ-HSD2 enzyme, and promote fetal lung maturation and preterm survival rate. (**C**) Increased plasma cortisol levels resulting from excess GC-induced perturbations to the hypothalamus–pituitary–adrenal axis may directly eliminate pre-leukemic cells and suppress leukemia-promoting Th1-cytokine responses. (**D**) Mechanisms of action of glucocorticoid in ALL therapy. Glucocorticoid-mediated apoptosis is thought to be induced via the mitochondrial pathway through caspase activation.

**Table 1 ijms-26-02971-t001:** Prevalence of the neonatal *ETV6::RUNX1* fusion in the intervention groups, including only participants with high adherence to the intervention.

Intervention Group	Neonatal *ETV6::RUNX1*+ No. (%)	Neonatal *ETV6::RUNX1*− No. (%)	OR (95% CI)	*p*-Value
Usual care	17 (7.0)	227 (93.0)	reference	NA
Stress reduction	8 (5.9)	128 (94.1)	0.8 (0.3–2.0)	0.683
Mediterranean Diet	9 (5.3)	162 (94.7)	0.7 (0.3–1.7)	0.482

**Table 2 ijms-26-02971-t002:** Univariate analysis of the association between maternal and prenatal characteristics with neonatal *ETV6-RUNX1* positivity.

	No. (%)		Crude Odds Ratio for Neonatal *ETV6::RUNX1*+	
	Neonatal *ETV6::RUNX1* Positive (*n* = 48)	Neonatal *ETV6::RUNX1* Negative (*n* = 693)	OR (95% CI)	*p*-Value
**Maternal Baseline Characteristics**				
Maternal age (years)	36.8 (5.0)	36.8 (5.1)	1.00 (0.95–1.06)	0.969
BMI before pregnancy (kg/m^2^)	23.6 (4.0)	24.1 (4.8)	0.97 (0.9–1.04)	0.427
Ethnicity				
White	34 (70.8)	544 (78.5)	Reference	Reference
Black	0 (0)	14 (2.0)	NA	NA
Asian	1 (2.1)	15 (2.2)	1.1 (0.1–8.3)	0.951
Indian	0 (0)	5 (0.7)	NA	NA
Latin American	9 (18.8)	110 (15.9)	1.3 (0.6–2.8)	0.489
**Maghreb**	**4 (8.3)**	**5 (0.7)**	**12.8 (3.3–49.9)**	**<0.001**
Study class				
Primary/no studies	5 (10.4)	43 (6.2)	Reference	Reference
**Secondary/technological**	**9 (18.7)**	**247 (35.6)**	**0.3 (0.1–0.98)**	**0.046**
University	34 (70.8)	403 (58.2)	0.7 (0.3–1.9)	0.525
Socioeconomic status ^ψ^				
Low	5 (10.4)	43 (6.2)	Reference	Reference
Medium	12 (25.0)	270 (39.0)	0.4 (0.1–1.1)	0.084
High	31 (64.6)	380 (54.8)	0.7 (0.3–1.9)	0.485
Nulliparous	27 (56.3)	399 (57.6)	0.95 (0.5–1.7)	0.857
Pregestational diabetes	5 (10.4)	38 (5.5)	2.0 (0.8–5.4)	0.165
Thyroid disorder	4 (8.3)	90 (13.0)	0.6 (0.2–1.7)	0.353
Autoimmune disease	6 (12.5)	112 (16.2)	0.7 (0.3–1.8)	0.504
Chronic hypertension	1 (2.1)	29 (4.2)	0.5 (0.1–3.7)	0.484
Chronic kidney disease	1 (2.1)	17 (2.5)	0.8 (0.1–6.5)	0.872
Obesity ^ф^	3 (6.3)	85 (12.3)	0.5 (0.1–1.6)	0.223
**Pregnancy and prenatal characteristics**				
Intervention group				
Usual care	17 (35.4)	228 (32.9)	Reference	Reference
Stress reduction	16 (33.3)	230 (33.3)	0.9 (0.5–1.9)	0.847
Mediterranean Diet	15 (31.3)	235 (33.9)	0.85 (0.4–1.8)	0.671
Cigarette smoking	2 (4.2)	54 (7.8)	0.5 (0.1–2.2)	0.367
Alcohol intake	0 (0)	17 (2.5)	NA	NA
Recreational drug consumption	0 (0)	3 (0.4)	NA	NA
Folate supplementation	31 (64.6)	532 (76.8)	0.6 (0.3–1.2)	0.059
**Exogenous corticosteroids**	**7 (14.6)**	**32 (4.6)**	**3.5 (1.5–8.5)**	**0.005**
Gestational diabetes	3 (6.3)	81 (11.7)	0.5 (0.2–1.7)	0.259
Preterm birth	4 (8.3)	37 (5.3)	1.6 (0.5–4.7)	0.386
Preeclampsia	0 (0)	57 (8.2)	NA	NA
Small for gestational age ^Υ^	7 (14.6)	27 (3.9)	0.8 (0.3–1.8)	0.591
Gestational age at delivery (weeks)	39.2 (1.9)	39.4 (1.7)	0.9 (0.8–1.1)	0.404
Neonatal sex				0.367
Female	20 (41.7)	335 (48.4)	Reference	
Male	28 (58.3)	357 (51.6)	1.3 (0.7–2.4)	

Results are displayed as n (%) or mean (SD). BMI denotes body mass index. ^ψ^ Socio economic status defined as low if participants reported having never worked or being unemployed for more than 2 years and having a partner with unqualified work or who was unemployed; high if they reported university studies regardless of whether they were working; and medium if any other situations. ^ф^ Obesity is defined as body mass index above 30. ^Υ^ Small for gestational age is defined as birthweight below the 10th centile according to local standards.

**Table 3 ijms-26-02971-t003:** Multivariate regression models for maternal and prenatal risk factors for neonatal ETV6::RUNX1 positivity.

	Model 1		Model 2		Model 3	
	Crude OR (95%CI)	*p*-Value	Adjusted OR (95%CI)	*p*-Value	Adjusted OR (95%CI)	*p*-Value
Maghreb ethnicity	12.8 (3.3–9.9)	<0.001	14.8 (3.4–63.9)	<0.001	10.5 (2.4–45.4)	0.002
Exogenous corticosteroids	3.5 (1.5–8.5)	0.005	3.4 (1.4–8.4)	0.007	3.9 (1.6–9.8)	0.003
Study class Primary/no studies Secondary/technological University	Reference 0.3 (0.1–0.98) 0.7 (0.3–1.9)	0.0367	Reference 0.5 (0.2–1.9) 1.4 (0.4–4.6)	0.03	Reference 0.5 (0.1–1.9) 1.4 (0.4–4.4)	0.306
Folate supplementation	0.6 (0.3–1.2)	0.059	0.6 (0.3–1.1)	0.10	0.6 (0.3–1.1)	0.083

Model 1 is unadjusted. Model 2 is adjusted for ethnicity and corticosteroids, study class, and folate supplementation. Model 3 is adjusted for variables in model 2 plus neonatal sex, pregestational diabetes, and smoking habit.

**Table 4 ijms-26-02971-t004:** Prevalence of neonatal *ETV6::RUNX1* positivity according to the class of exogenous corticosteroid.

Exogenous Corticosteroid Class	Neonatal *ETV6::RUNX1*+ (*n* = 7)	Neonatal *ETV6::RUNX1*− (*n* = 32)	OR (95% CI)	*p*-Value
Betamethasone (GR:MR = 25:1)	5	20	4.0 (1.4–11.3)	0.008
Metilprednisone (GR:MR = 11:1)	1	0	NA	NA
Prednisone (GR:MR = 5:1)	1	10	1.6 (0.2–12.9)	0.653
Topical corticosteroids (GR:MR = 10–100:1, according to steroid class)	0	2	NA	NA

CI denotes confidence interval; NA, not applicable. GR:MR refers to the glucocorticoid receptor (GR) to mineralocorticoid receptor (MR) activity ratio for each exogenous corticosteroid listed.

## Data Availability

The data presented in this study are available on request from the corresponding author due to local IRB requirements.

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
