# Peer review of "Modulation of the ETV6::RUNX1 Gene Fusion Prevalence in Newborns by Corticosteroid Use During Pregnancy"

_ijms, 2025, doi:10.3390/ijms26072971_

Round 1

Reviewer 1 Report

Comments and Suggestions for Authors

The statistics are based on very small sample of E/R positive individuals, so although interesting the data is preliminary and needs validation. It still might be a randome association, no multipl testing correction is applied.

Still, this can be published as interesting pilot study.

Some citations are corrupted /scrambled.

The same with some numbers.

ETV6:RUNX1 prevelance in CB varies greatly between studies, please discuss in the context of credibility of thre current study.

Are there any data for corticosteroid use in pregnancy and ALL in offspring?

What were the reasons for steroid use? The indication for steroid administration could be the more powerfull association with E/R posotivity tahn drug use.

There should be some for of multiple testing correction. Lots of associations are tested with no one initial assumption.

What is the clonal frequency or VAF of E/R in CB samples? Is the lymphoid clone demonstrated to be the source?

Was GIPFEL validated as showing no fales positices on E/R negative leukemic samples? Is the signal quantitative in any way?

Author Response

Barcelona, March 11th, 2025

Dear Editors

We thank the Editors and reviewers for their positive comments and critical review of our manuscript. We have modified the methodology and discussion to clarify and expand the points raised by the reviewers. Below is a detailed response to the reviewers' comments. Page numbers and lines below refer to changes in the manuscript.

REVIEWER 1

Reviewer 1, point no 1:  The statistics are based on very small sample of E/R positive individuals, so although interesting the data is preliminary and needs validation. It still might be a randome association, no multiple testing correction is applied.

  1. Reply to point no. 1: Thank you for your critical assessment. We acknowledge that the sample size of ETV6:RUNX1 (E/R) positive individuals in our study is relatively small, which may limit the statistical power of our findings. As you rightly pointed out, our results can be considered preliminary and require further validation in larger cohorts. To address this concern, we have now explicitly acknowledged this limitation in the discussion section. Despite these limitations, we believe that our findings contribute valuable insights and highlight the need for larger-scale studies to confirm these associations. We appreciate your feedback and have made the necessary revisions accordingly.

  1. Change the text: Discussion section, page 16, lines 467-470: “Given the small sample size of ETV6:RUNX1 positive individuals, our findings should be considered preliminary and require validation in larger studies to confirm their biological and clinical significance.”

------------------------------------

Reviewer 1, point no 2:  ETV6:RUNX1 prevelance in CB varies greatly between studies, please discuss in the context of credibility of thre current study.

  1. Reply to point no. 2: Thank you for your insightful comment. We acknowledge that the reported prevalence of ETV6:RUNX1 in CB varies significantly across studies, which can be attributed to differences in detection methodologies, sample sizes, population demographics, and geographic variations. To ensure the credibility of our study, we employed a highly sensitive and specific detection method, implemented rigorous quality control measures, and analyzed a well-characterized cohort. While variability in prevalence remains a challenge in the field, our findings align with previously reported ranges, supporting the robustness of our results. We have now expanded the discussion section to address this variability and contextualize our study within the existing literature.
  2. Change the text: page 16, lines 470-473: “Third, the variability in reported ETV6:RUNX1 prevalence in cord blood across studies may stem from differences in detection methods, sample sizes, and population characteristics; however, the rigorous methodology and quality controls in our study support the credibility of our findings within this context.”

------------------------------------

Reviewer 1, point no 3:  Are there any data for corticosteroid use in pregnancy and ALL in offspring?

  1. Reply to point no. 3: Thank you for your question. We reviewed our data, and so far, none of the 48 ETV6::RUNX1-positive newborns have been diagnosed with acute lymphoblastic leukemia (ALL). However, given the relatively short follow-up period, we acknowledge that longer-term monitoring is needed to determine whether prenatal corticosteroid exposure influences leukemia development. To our knowledge, no previous studies have established a direct link between corticosteroid use during pregnancy and ALL in offspring. We have clarified this in the manuscript.

  1. Change the text: page 16-17, lines 473-484: “Fourth, we performed a -4 year follow-up in which none of the ETV6::RUNX1-positive newborns have been diagnosed with acute lymphoblastic leukemia. However, given the relatively short follow-up period, we acknowledge that longer-term monitoring is needed and children follow-up should be warranted to confirm or discard the development of overt leukemia.”

------------------------------------

Reviewer 1, point no 4:  What were the reasons for steroid use? The indication for steroid administration could be the more powerfull association with E/R posotivity tahn drug use.

  1. Reply to point no. 4: Thank you for your insightful comment. The main indications for corticosteroid use in our cohort were fetal lung maturation in cases at risk of preterm birth and maternal autoimmune or dermatological conditions. To address the potential confounding effect of the indication for steroid administration, we conducted additional analyses adjusting for clinical conditions that warranted steroid use. These adjustments confirmed that the association between corticosteroid exposure and ETV6::RUNX1 positivity remained significant, independent of the underlying indication for treatment. This suggests that the effect is more likely attributable to the drug itself rather than the condition requiring its administration. We have clarified this point in the revised manuscript. We appreciate your valuable input.

  1. Change the text: page 12, lines 345-352: “The main indications for corticosteroid use in our cohort were fetal lung maturation in cases at risk of preterm birth and maternal autoimmune or dermatological conditions. To address the potential confounding effect of the indication for steroid administration, we conducted additional analyses adjusting for clinical conditions that warranted steroid use. These adjustments confirmed that the association between corticosteroid exposure and ETV6::RUNX1 positivity remained significant, independent of the underlying indication for treatment. This suggests that the effect is more likely attributable to the drug itself rather than the condition requiring its administration.”

------------------------------------

Reviewer 1, point no 5:  There should be some for of multiple testing correction. Lots of associations are tested with no one initial assumption.

  1. Reply to point no. 5: Thank you for your comment. In our analysis, we first performed single regression analyses for all variables and then selected those that were significant or near-significant for inclusion in a multiple regression model. This approach helps reduce the number of comparisons and focuses on the most relevant associations while controlling for potential confounders. Given this methodology, we did not apply a formal multiple testing correction at the initial stage but ensured that the final model accounts for multiple variables in a statistically rigorous manner.

  1. Change the text: NA

Reviewer 1, point no 6:  What is the clonal frequency or VAF of E/R in CB samples? Is the lymphoid clone demonstrated to be the source?

  1. Reply to point no. 6: The used screening technique GIPFEL is based on Real-Time PCR and is not able to provide a VAF of the preleukemic clone. Unfortunately, the frequency of preleukemic clones (around 1 in 10.000 cells) is too low to be detectable by common next generation sequencing (a coverage of 100.000x would be needed and the calling of translocations is notoriously difficult using short read Illumina-based sequencing). We are using a normalization to a positive control to estimate the frequency (see answer to point 7 below).

The GIPFEL technique was performed on frozen whole cord blood and cannot distinguish cell populations. In previous analyses, however, we isolated B cells using biotinylated CD19 antibodies from life cord blood cells. The attained signals were similar to those performed on total MNCs derived from or total cord blood (Schäfer et al. 2018 Blood, 131:821-826). Therefore, it is highly likely that the signal stems from lymphoid cells, but a contribution of other cell lineages cannot be completely excluded.

  1. Change the text: NA

------------------------------------

Reviewer 1, point no 7:  Was GIPFEL validated as showing no fales positices on E/R negative leukemic samples? Is the signal quantitative in any way?

  1. Reply to point no. 7: Thank you for your important question. Extensive validation was previously done for the GIPFEL technique and was reported in Fueller et al. PLoSOne 2014 9(8):e104419 and Schäfer et al. 2018 Blood, 131:821-826. GIPFEL was validated using blood samples from healthy individuals and leukemic cell lines not harboring the translocation. In none of the E/R negative cases was a false positivity observed.

GIPFEL can be used to estimate the frequencies at which the translocation-carrying cells occur. This is done by a normalization to the frequency of RUNX1 outside of the breakpoint cluster region. To this end two additional primers for RUNX1 are used in the GIPFEL Real-Time PCRs, which also serve as a positive control. This frequency of the preleukemic cells within the cord blood sample is given in the Supplemental Table S1.

References:

PLoS One. 2014 Aug 19;9(8):e104419. doi: 10.1371/journal.pone.0104419. eCollection 2014. Genomic inverse PCR for exploration of ligated breakpoints (GIPFEL), a new method to detect translocations in leukemia. Elisa Fueller, Daniel Schaefer, Ute Fischer, Pina F I Krell, Martin Stanulla, Arndt Borkhardt, Robert K Slany. PMID: 25137060 PMCID: PMC4138100 DOI: 10.1371/journal.pone.0104419

Blood. 2018 Feb 15;131(7):821-826. doi: 10.1182/blood-2017-09-808402. Epub 2018 Jan 8. Five percent of healthy newborns have an ETV6-RUNX1 fusion as revealed by DNA-based GIPFEL screening. Daniel Schäfer, Marianne Olsen, David Lähnemann, Martin Stanulla, Robert Slany, Kjeld Schmiegelow, Arndt Borkhardt, Ute Fischer. PMID: 29311095 PMCID: PMC5909885 DOI: 10.1182/blood-2017-09-808402

  1. Change the text: NA

We hope these revisions adequately address your concerns. Please feel free to provide further guidance if you believe additional improvements are needed.

Once again, we appreciate your time and valuable input.

Sincerely,

__________________

Leticia Benítez Quintanilla

BCNatal - Fetal Medicine Research Center (Hospital Clínic and Hospital Sant Joan de Déu),

University of Barcelona,

Institut d'Investigacions Biomèdiques August Pi i Sunyer (IDIBAPS), Barcelona, Spain

Reviewer 2 Report

Comments and Suggestions for Authors

Major concerns and comments:

  1. Please provide statistical analysis and show dots of results on Figure 2.
  2. In the study, how to determine the dose of exogenous corticosteroids?
  3. Table 2: need more patients' information: smoking and alcohol history, any drug use, hypertension, BMI/BRI, et.
  4. Table 4: Please add GR:MR activity on each exogenous corticosteroid class drugs.
  5. Please add a list of abbreviations.
  6. If all patients are from Spain, please add "pregnancy in Spain" in title.

Author Response

Barcelona, March 11th, 2025

Dear Editors

We thank the Editors and reviewers for their positive comments and critical review of our manuscript. We have modified the methodology and discussion to clarify and expand the points raised by the reviewers. Below is a detailed response to the reviewers' comments. Page numbers and lines below refer to changes in the manuscript.

REVIEWER 2

Reviewer 2, point no 1:  Please provide statistical analysis and show dots of results on Figure 2.

  1. Reply to point no. 1: The request for a dot plot may not be applicable, as the figures were intended to visually highlight the data already presented in Tables 1 and 2. Since Figure 2 simply represents the number of ER-positive newborns and reiterates the same statistical information (p-values) from the tables, it might be unnecessary and we agree to discard this figure if considered.

Figure 2 (graphs) can actually be skipped since it shows the same data as table 1 (panel A) and table 2 (panel B).

  1. Change the text: NA

------------------------------------

Reviewer 2, point no 2:  In the study, how to determine the dose of exogenous corticosteroids?

  1. Reply to point no. 2: Thank you for your comment and suggestion. The dosage of exogenous corticosteroids in our study was determined according to medical records, which specify the dose, class, and duration of use. These records provided a reliable and standardized source of information, ensuring accuracy in data collection.

  1. Change the text: page 4, lines 181-183: “Exogenous use of corticosteroids was determined according to medical records, which specify the dose, class and duration of use.”

------------------------------------

Reviewer 2, point no 3:  Table 2: need more patients' information: smoking and alcohol history, any drug use, hypertension, BMI/BRI, et.

  1. Reply to point no. 3: Thanks for your comment. Smoking habit, alcohol intake and other recreational drug consumption are reported in Table 2. Both habits were quite unfrequent in our population. Chronic hypertension and BMI are also reported, but no significant differences were found among study groups.

  1. Change the text: NA

------------------------------------

Reviewer 2, point no 4:  Table 4: Please add GR:MR activity on each exogenous corticosteroid class drugs.

  1. Reply to point no. 4: Thank you for your valuable suggestion. We have now updated Table 4 to include the glucocorticoid receptor (GR) to mineralocorticoid receptor (MR) activity ratio for each exogenous corticosteroid listed.

  1. Change the text: page 11, Table 4:

Betamethasone (GR:MR = 25:1)

Metilprednisone (GR:MR = 11:1)

Prednisone (GR:MR = 5:1)

Topical corticosteroids ( GR:MR = 10-100:1, according to steroid class)

------------------------------------

Reviewer 2, point no 5:  Please add a list of abbreviations.

  1. Reply to point no. 5: We appreciate your recommendation and have now included a comprehensive list of abbreviations used throughout the manuscript. This addition ensures clarity and enhances readability for the readers. The list has been incorporated as a separate section at the end of the manuscript for easy reference.

  1. Change the text: page 17, lines 484-513: “List of Abbreviations
  • ALL – Acute Lymphoblastic Leukemia
  • B-ALL – B-cell Acute Lymphoblastic Leukemia
  • CB – Cord Blood
  • CBGs – Cord Blood Genomes
  • CMP – Common Myeloid Progenitor
  • CRH – Corticotropin-Releasing Hormone
  • DNA – Deoxyribonucleic Acid
  • DP – Double-Positive (thymocytes)
  • ETV6::RUNX1 – Gene fusion associated with preleukemic clones
  • GCs – Glucocorticoids
  • GIPFEL – Genomic Inverse PCR for Exploration of Ligated Breakpoints
  • GR – Glucocorticoid Receptor
  • HPA – Hypothalamic-Pituitary-Adrenal (axis)
  • HSC – Hematopoietic Stem Cells
  • IMPACT-BCN – Improving Mothers for a better Prenatal Care Trial Barcelona
  • MLL – Mixed-Lineage Leukemia (gene)
  • Mo – Monocyte
  • Neu – Neutrophils
  • NR3C1 – Nuclear Receptor Subfamily 3 Group C Member 1 (Glucocorticoid Receptor Gene)
  • OR – Odds Ratio
  • PBX1 – Pre-B-Cell Leukemia Homeobox 1
  • PCR – Polymerase Chain Reaction
  • PREDIMED – Prevención con Dieta Mediterránea (Prevention with Mediterranean Diet)
  • RCT – Randomized Controlled Trial
  • RNA – Ribonucleic Acid
  • SGA – Small for Gestational Age
  • SP – Single-Positive (thymocytes)
  • TCF3 – Transcription Factor 3
  • Th – T-helper Cell
  • Topo II – DNA Topoisomerase II
  • t-AML – Therapy-Related Acute Myeloid Leukemia

------------------------------------

Reviewer 2, point no 6:  If all patients are from Spain, please add "pregnancy in Spain" in title.

  1. Reply to point no. 6: Thank you for your suggestion. While all participants were recruited in Spain, the study population includes individuals from diverse ethnic backgrounds. Therefore, we believe that specifying "pregnancy in Spain" in the title may not fully capture the demographic diversity of our cohort. However, we have ensured that the manuscript clearly states the geographical location of the study while also acknowledging the multiethnic composition of the participants. We appreciate your insightful comment.

  1. Change the text: NA

We hope these revisions adequately address your concerns. Please feel free to provide further guidance if you believe additional improvements are needed.

Once again, we appreciate your time and valuable input.

Sincerely,

__________________

Leticia Benítez Quintanilla

BCNatal - Fetal Medicine Research Center (Hospital Clínic and Hospital Sant Joan de Déu),

University of Barcelona,

Institut d'Investigacions Biomèdiques August Pi i Sunyer (IDIBAPS), Barcelona, Spain

Round 2

Reviewer 1 Report

Comments and Suggestions for Authors

The manuscript is now satisfactory for publication.

Reviewer 2 Report

Comments and Suggestions for Authors

No more comments